# Development of a core outcome set for multimorbidity trials in low/middle-income countries (COSMOS): study protocol

Jan R. Boehnke [1,2] Rusham Zahra Rana,[3] Jamie J. Kirkham [4]
Louise Rose [5] Gina Agarwal [6] Corrado Barbui,[7,8] Alyssa Chase-Vilchez,[9]
Rachel Churchill,[10] Oscar Flores-Flores,[11,12] John R. Hurst [13] Naomi Levitt,[14]
Josefien van Olmen,[15] Marianna Purgato,[7,8] Kamran Siddiqi [2,16]
Eleonora Uphoff,[10] Rajesh Vedanthan [17] Judy Wright [18] Kath Wright,[19]
Gerardo A. Zavala,[2] Najma Siddiqi [2,16]

For numbered affiliations see end of article.

**Correspondence to**
Dr Jan R. Boehnke;
j.r.boehnke@dundee.ac.uk

## ABSTRACT

**Introduction** 'Multimorbidity' describes the presence of two or more long-term conditions, which can include communicable, non-communicable diseases, and mental disorders. The rising global burden from multimorbidity is well documented, but trial evidence for effective interventions in low-/middle-income countries (LMICs) is limited. Selection of appropriate outcomes is fundamental to trial design to ensure cross-study comparability, but there is currently no agreement on a core outcome set (COS) to include in trials investigating multimorbidity specifically in LMICs. Our aim is to develop international consensus on two COSs for trials of interventions to prevent and treat multimorbidity in LMIC settings.

**Methods and analysis** Following methods recommended by the Core Outcome Measures in Effectiveness Trials initiative, the development of these two COSs will occur in parallel in three stages: (1) generation of a long list of potential outcomes for inclusion; (2) two-round online Delphi surveys and (3) consensus meetings. First, to generate an initial list of outcomes, we will conduct a systematic review of multimorbidity intervention and prevention trials and interviews with people living with multimorbidity and their caregivers in LMICs. Outcomes will be classified using an outcome taxonomy. Two-round Delphi surveys will be used to elicit importance scores for these outcomes from people living with multimorbidity, caregivers, healthcare professionals, policy makers and researchers in LMICs. Finally, consensus meetings including all of these stakeholders will be held to agree outcomes for inclusion in the two COSs.

**Ethics and dissemination** The study has been approved by the Research Governance Committee of the Department of Health Sciences, University of York, UK (HSRGC/2020/409/D:COSMOS). Each participating country/research group will obtain local ethics board approval. Informed consent will be obtained from all participants. We will disseminate findings through peer-reviewed open access publications, and presentations at global conferences selected to reach a wide range of LMIC stakeholders.

## Strengths and limitations of this study

► The development process follows guidelines and best practice recommendations for developing core outcome sets and integrates four sources of information.

► Interviews with people living with multimorbidity and caregivers are conducted in several low-/middle-income countries (LMICs) (in South Asia, Africa and Latin America), by local teams, and in local languages to identify outcomes relevant to them.

► The Delphi survey and consensus meetings are conducted in English which limits the breadth of participation in these stages of the process.

► Despite involvement of a wide range of LMIC stakeholders in the process, there may be some limitations to the generalisability of the final core outcome sets due to the heterogeneity of target conditions and the diversity of countries, cultures, and experiences.

**PROSPERO registation number** CRD42020197293.

## INTRODUCTION

The rising global burden from chronic diseases is widely recognised and its detrimental impact on individuals as well as on societies is well documented.[1 2] Chronic diseases often co-occur in individuals and this 'multimorbidity' is associated with more frequent healthcare consultations,[3] longer hospital stays,[4] worse health-related quality of life,[5–7] increased healthcare costs,[8] higher out-of-pocket expenditure for healthcare[8] and higher mortality.[9] These factors make multimorbidity a pressing challenge for healthcare systems as well as for patients, families and caregivers.[2 8 10] We define multimorbidity as the presence of two or more chronic (or

long-term) health conditions[11] encompassing communicable and non-communicable diseases and mental disorders, for example, depression and diabetes occurring together. Chronic communicable diseases are those that affect people over a number of months or years, for example, HIV or tuberculosis. Non-communicable diseases are chronic conditions that cannot be directly transmitted between people, for example, diabetes, heart disease, and chronic lung disease.

Estimates of the prevalence of multimorbidity in high-income countries (HICs) vary from 25% to 60% in healthcare and community settings.[12–14] The prevalence of multimorbidity in low-/middle-income countries (LMICs) appears to be lower but is set to rise rapidly due to demographic and lifestyle changes.[11 15 16] A Cochrane review on the effectiveness of health service or patient-oriented interventions to improve outcomes in people with multimorbidity in primary care and community settings found most evidence was from HICs.[17] This evidence cannot readily be applied to LMICs where differing patterns of disease, healthcare systems, resources and cultural considerations will affect the acceptability and effectiveness of interventions.[11 18] Trials of interventions to tackle multimorbidity specifically in LMICs are urgently needed.[10]

To generate a robust evidence-base, results from individual trials will need to be compared and synthesised. This requires studies to use a core outcome set (COS), that is, an agreed minimum set of outcomes to be measured in all trials of a specific intervention for a health condition.[19] A COS for multimorbidity research has been previously developed[20]; it includes three outcomes: health-related quality of life, mental health and mortality. However, a limitation of this COS is that it was largely based on the evidence and expert opinions from HICs, with no involvement of LMIC stakeholders in reaching consensus. This is a major barrier to its adoption in LMIC research, where there may be differing priorities for treatments among patients, families, healthcare professionals, and researchers, and therefore a different set of core outcomes may be required, specifically for LMIC settings.[21 22] A recent review discussed outcome measures across eight domains that may be suitable for multimorbidity research in LMICs.[10] However, it did not systematically review the literature, patients and caregivers were not involved, and outcomes were not prioritised using established consensus methods. The authors concluded that development of a COS for LMICs would be an important contribution to the field.

While some research outcomes may be applicable to both LMICs and HICs, there are important differences in disease patterns and healthcare systems that need to be considered. For example, differences in severity of disease symptoms or complications and in healthcare availability and financing may influence outcome priorities.[1 15 16 21] In a recent priority setting exercise for trials in LMIC settings, the choice of outcomes to measure was identified as the number one most important topic.[23] This protocol outlines a plan to bridge the gap and develop two COSs for research on the prevention and treatment of multimorbidity in an LMIC setting, using the minimum standards approach for developing COS.[24] Separate COSs are needed given the often differing targets for prevention and treatment interventions.

## Aims and objectives

We aim to develop two COSs for trials of interventions designed to (1) prevent and (2) treat multimorbidity in LMIC settings. The scope includes all types of interventions for multimorbidity — pharmacological, non-pharmacological, simple and complex, in adults (aged 18 years and over) at risk of, or living with multimorbidity, in community, primary care, and hospital settings in LMICs. Complex interventions are those with multiple interacting components (eg, health risk behaviour change interventions), which contrast with 'simple' interventions, such as pharmacological interventions.[25]

## METHODS

We will follow approaches recommended by the Core Outcome Measures in Effectiveness Trials (COMET) initiative[26] and Outcome Measures in Rheumatoid Arthritis Clinical Trials.[27] The COS development has been registered with COMET (https://www.comet-initiative.org/Studies/Details/1580). The process will follow the minimum standards for the design of a COS study (COS-STAD[24]), which includes the careful consideration of the scope, stakeholders and the consensus process. This protocol is structured according to the COS-STAP statement.[28]

Our research team has broad global LMIC representation as well as experience of working in LMICs. The Core Outcome Set for Multimorbidity Studies (COSMOS) working group supporting the study draws on a network of 38 research teams in LMICs that will be actively involved in different stages of COS development. We will ensure patient, community and public involvement throughout the development and delivery of the project, and include people with lived experience and their caregivers as members in the core research team.

COS development will involve three stages: (1) generation of a long list of outcomes, (2) two-round online Delphi consensus-building surveys, and (3) online consensus meetings.

## Outcome list generation

A long-list of outcomes will be generated from two sources: first, a systematic review of outcomes reported in trials of interventions conducted in LMICs for the prevention and treatment of multimorbidity, and second, using interviews with people living with multimorbidity and their caregivers from LMICs.

### Systematic review

We will conduct a systematic review to identify an initial list of potential outcomes from trials of multimorbidity

interventions conducted in adults living in LMICs. The systematic review protocol has been registered, and full details are available on PROSPERO.

### Search methods

We will search electronic databases (eg, MEDLINE, Embase, PsycINFO) from 1990 to present. We will also search for relevant systematic reviews in The Cochrane Library, the Database of Abstracts of Reviews of Effects, PROSPERO and Joanna Briggs; and unpublished ongoing trials in ClinicalTrials.gov, ISRCTN registry and in the International Clinical Trials Registry Platform (http://apps.who.int/trialsearch). We will apply a study design filter to identify interventional studies. We will not limit inclusion by language.

### Selection criteria

We will include randomised (individual, cluster and cross-over randomisation) intervention studies published in either protocols or definitive reports. Studies of adults aged 18 years and above, living in LMICs, with multimorbidity or at risk of multimorbidity will be eligible. LMICs will be defined using the 2019 World Bank definitions.[29] Studies of all interventions, whether pharmacological, non-pharmacological, simple or complex, for the prevention or treatment of multimorbidity (or both) and any comparators will be included.

### Study selection

Two reviewers will independently screen titles/abstracts with any discrepancies resolved through consensus or if needed, by a third reviewer. Full texts will also be reviewed following a similar process.

### Data collection

Data will be extracted from eligible studies on publication date, years when the trial was conducted, country, clinical setting (eg, primary care, specialised services), type of intervention, target conditions and outcomes, by a single reviewer, with 10% of extraction records also reviewed by a second senior reviewer.

### Quality assessment

Risk of bias will not be assessed, as the purpose is only to assemble a list of outcomes that have been used in previous studies.

### Data synthesis

Outcomes will be grouped based on whether they were used to evaluate prevention or treatment of multimorbidity. For both groups of outcomes, we will construct matrices to identify the outcomes, whether primary or secondary, their definitions and measures used in each study. We will tabulate the proportion of included studies that report on each outcome (including primary vs other outcome) and rank-order the outcomes accordingly. In a second step, to organise the outcomes, we will use an outcome taxonomy.

The outcomes will be presented by intervention type (prevention, treatment), by combination of conditions that the trial targeted, and by region of the world (World Bank regions of LMICs[29]) in which the study was conducted.

### Interviews

To identify outcomes of importance to people from LMICs living with multimorbidity and their families or caregivers, we will conduct semistructured interviews. Participants will be approached by local research teams either in-person or via phone, post or online (based on the local research site resources and COVID-19 restrictions).

Inclusion criteria for interviews comprise: people living with (or caring for someone living with) two or more health conditions; at least 18 years of age; being able to visit the local partner organisation's site or access to telephone/internet and consenting to participate. Participants will be identified through contact with relevant healthcare services, support and advocacy groups, charities, and use of social media including Twitter, adverts placed on public and patient involvement websites, snowballing techniques and personal contacts. We will use purposive sampling[30] based on the following characteristics: age (over/under 65 years); sex (male/female); using community or primary care versus secondary/specialist healthcare services, and region (World Bank income groups: low, lower middle, upper middle[29]). We will include a range of LMICs in diverse geographic locations. We anticipate 15–20 participants from each group (45–60 interviews), which should be sufficient to achieve diversity of viewpoints and data saturation.[31]

We will collect demographic information (age, sex, marital status, highest level of education, socioeconomic status, occupation, and disability). We have developed a semistructured interview guide (see online supplemental appendix 1),[26] which will be translated and back-translated by local teams in the local language. Interviews will be conducted by researchers with experience of conducting qualitative interviews. All interviews will be audio recorded or if not possible (because of technology limitations or the participant withholding consent) will be recorded in contemporaneous notes.

The local research teams will be asked to transcribe and translate into English two sections of the interview, (1) participants' consent and (2) content relevant to study outcomes. Interview transcripts will be reviewed to confirm health outcomes to put forward for the Delphi.

Outcomes identified in the interviews will be integrated with findings from the systematic review. All unique outcomes identified from the systematic review and interviews will be categorised into domains for presentation in the Delphi survey using Dodd's outcome taxonomy.[32]

### Delphi surveys with stakeholders

We will conduct separate Delphi surveys for the prevention and treatment COS; however, survey participants will be common to both.

We will recruit a range of stakeholders living or working in LMICs including the following groups: (1) people living with multimorbidity and their family caregivers, (2) healthcare professionals experienced in working with patients with multimorbidity; (3) policy makers and managers and (4) researchers interested in multimorbidity with experience in LMIC contexts. Participants will be identified using connected networks (eg, National Institute for Health Research Global Health Research Groups/Units, Global Alliance for Chronic Diseases, World Psychiatric Association, NCD Alliance) and our project network across 38 countries to distribute information about the study, including professional societies and non-government organisations relevant to multimorbidity and government ministries. For recruitment to the researcher stakeholder group, we will send personalised emails to corresponding authors living or working in LMICs identified via our systematic review and request them to snowball to co-authors and other relevant contacts.

We will aim to maintain a minimum of approximately 20 participants representing each stakeholder group (total approximately 80 participants) throughout Delphi rounds, with representation across the World Bank LMIC income groups.[29] We will oversample for the first round based on an estimated attrition of 30% across rounds. In both rounds, we will send three reminders across 3 weeks to participants to complete the surveys and keep the surveys open for longer if needed to reduce attrition and improve the response rate.

### Round one

We will include all outcomes identified through our systematic review and interviews, seeking advice from our community and public representatives to describe outcomes in lay terms. We will ensure the outcomes that enter the Delphi are non-disease specific (important for the scope of this COS), have relevance to LMICs, and avoid duplication. We will use an online survey tool (DelphiManager) to administer the surveys. We will pilot the questionnaire with eight individuals (two from each stakeholder group) to assess content validity and understanding.

We will list outcomes in domains according to the taxonomy classification. Since we are interested in assessing the importance of each outcome, we will ask participants to score each outcome (separately for prevention and for treatment) on a scale from 1 to 9 (1–3=not important for inclusion; 4–6=important but not critical; 7–9=critical for inclusion[33]), considering relevance for and feasibility of the outcome in LMIC settings. There will also be an option for 'unable to score'. We will provide the opportunity for participants to add additional outcomes and other comments. We will collect demographic information to describe our study sample (table 1). All responses, including from partially completed questionnaires, will be used in analyses.

**Table 1** Delphi stakeholder characteristics

| Stakeholder group | Characteristic |
|---|---|
| People with multimorbidity/family members/caregivers | Age (≤65; >65) |
| | Sex (male; female) |
| | Multimorbidities |
| | Community/primary care, secondary care; specialist healthcare |
| | Level of formal education (<=10 years/>10 years) |
| | Region of residence |
| Health professionals | Profession (doctor, nurse, allied health, community workers) |
| | Specialist area relevant to multimorbidity |
| | Region of residence |
| Policy makers and managers | Region of residence |
| Researchers | Specialist area relevant to multimorbidity |
| | Region of residence |

### Round two

The research team will review any additional outcomes suggested in round one for inclusion. Any newly suggested outcomes will be taken forward if they are (1) relevant to the scope of the COS and (2) not yet represented in the list of outcomes. All outcomes from round one and any new outcomes will be included in round two.

In round two, we will provide participants with their own round one response, summarised responses of the whole group, and responses summarised by each stakeholder group. We will ask them to re-score the importance of each outcome in light of the extra information received, again separately for prevention and treatment. New outcomes from round one will be provided for scoring on the same 1–9 importance scale.

We will summarise round two responses separately for prevention and treatment using descriptive statistics, noting for each outcome the number and percentage of responses in each of the following categories:
► 1–3=not important for inclusion.
► 4–6=important but not critical.
► 7–9=critical for inclusion.

We will categorise outcomes according to the predefined consensus definition described in table 2.[34–36] All responses, including from partially completed questionnaires will be used in analyses.

We will also examine the percentage distribution of scores for each outcome by stakeholder group, geographic region, and World Bank income group and present these visually for the consensus meeting.

At the end of the second round of Delphi surveys, participants will be sent information about the consensus

**Table 2** Definition of consensus for outcomes in the Delphi surveys

| Consensus IN | Consensus that outcome should be included in the core outcome set | 70% or more participants scoring 7–9 AND <15% participants scoring 1–3 |
|---|---|---|
| Consensus OUT | Consensus that outcome should not be included in the core outcomes set | 50% or fewer participants scoring 7–9 in each stakeholder group |
| No consensus | Uncertainty about importance of outcome | Anything else |

meeting and requested to contact the team if they are interested in participating.

### Outcome consensus meetings

Due to the wide geographic distribution of participants, the final step in the consensus process will be online consensus meetings to agree (1) the COS for prevention of multimorbidity and (2) the COS for treatment of multimorbidity.

Participants of the Delphi surveys who complete both rounds and who express an interest in attending the consensus meeting will be purposively selected to ensure similar numbers from each stakeholder group, geographical spread, and representation from target World Bank income groups. The meetings will be chaired by a facilitator experienced in COS development and will use a range of functions available in online meetings, such as breakout rooms for smaller group discussions, polls and 'chat' to facilitate meaningful participation. Results of the two-round Delphi surveys will be presented. Outcomes that have reached consensus 'in' or 'no consensus' (see table 2) will be discussed further.

We will use a modified nominal group technique to achieve consensus. This will involve iterative small and whole group discussions and ranking of outcomes in order to agree which should be included in the two COSs. To avoid duplication or overlap, the selected outcomes will also be discussed to ensure each relates to a distinct construct. If consensus cannot be reached on all outcomes or there are outstanding issues to consider, we will convene separate meetings with the stakeholder members until a final COS is ratified.

For people living with multimorbidity and their caregivers, we will provide additional support before the consensus meetings to explain the process and enable them to participate more fully in the discussions and ranking.

### PATIENT AND PUBLIC INVOLVEMENT

There were no people living with the condition nor their caregivers involved in the planning of the study. Both will be involved in the project management group overseeing the Delphi surveys and consensus meeting. Both will participate in both Delphi surveys and consensus meetings.

### ETHICS

The Ethics committee of the University of York has approved the study (HSRGC/2020/409/D: COSMOS). Approvals will also be sought from relevant local ethics committees for all participating sites.

We will obtain written consent from participants in interviews and consensus meetings (or where conducted online, verbal consent will be recorded). For the Delphi surveys, we will seek consent for participation as part of the online questionnaire and consent to store contact data for completion of the next Delphi round and for invitation to consensus meetings. We will emphasise that participation is voluntary, participants are free to withdraw at any stage and survey responses will be anonymised.

### Data management

Interview data as well as anonymised Delphi data after secure download will be stored on a secure and encrypted database on a secure password protected server (University of York, Department of Health Sciences File store Area). The access, use and storage of sensitive or confidential data will be conducted in accordance with the University of York Data Security Policy and Handling Sensitive Data Guidance.

### DISSEMINATION

We will disseminate our findings through peer-reviewed and open access publications, and presentations at global conferences, selected to reach a wide range of LMIC stakeholders including health professionals and researchers and taking into account geographic locations. We will also provide lay summaries including infographics in relevant languages for community and public audiences and share these with interview and Delphi participants. Such summaries and short briefing papers will be made available freely on the COSMOS web page (https://www.impactsouthasia.com/) and in formats which are at the moment being successfully developed and trialled in the associated research networks (Global Alliance for Chronic Diseases website, World Psychiatric Association). In the longer term, success of dissemination will be assessed by how frequently the multimorbidity COS is used in trials and cited in manuscripts.[37]

### DISCUSSION

We aim to develop and report two COSs for multimorbidity outcomes specifically for LMICs by identifying outcomes that have been reported in the literature, including the views of people living with multimorbidity and their caregivers through interviews, and developing

consensus using Delphi surveys and consensus meetings. Development of the two COSs aims to increase the consistency of selection, collection, and reporting of outcomes in future studies of interventions to prevent and to treat multimorbidity.

A considerable strength of the study is the involvement of people living with multimorbidity and their family caregivers throughout, including in interviews to gain their perspective on important outcomes. This involvement is increasingly recognised as an important contributor to COS[24 38 39] as it highlights new outcomes from the service user perspective. Aiming for approximately 50 participants from interviews and a wide range of participants in the Delphi globally,[39] the study will generate outcomes and gain consensus from diverse settings.

The research team comprises experienced researchers and experts in the field of multimorbidity and COS development. The support of the Global Alliance for Chronic Disease, the World Psychiatry Association and partners in IMPACT (Improving Outcomes in Mental and Physical Multimorbidity and Developing Research Capacity) will help provide links to local groups and identify appropriate stakeholders for the consensus process as well as for the dissemination of the outcomes generated from the study.

There are some relevant limitations. First, the scope is broad; despite the systematic effort to conduct a comprehensive and inclusive COS development, with involvement of a wide range of LMIC stakeholders, it is possible that there may be limitations to the generalisability of the final COSs due to heterogeneity of interventions, target conditions, and diversity of countries, cultures, and experiences.

A further limitation is that although the study aims to develop COS that will be relevant to LMICs more generally, it will only include representation from some LMIC regions. We recognise that LMICs are not a homogenous category and social and cultural differences will necessitate future research investigating how these COSs can be applied to other regions. Nevertheless, we included literature from all LMICs in generating the list of outcomes, and the diverse range of countries represented means it is likely that the outcomes selected will be those common across countries and cultures. We do not therefore intend to define the regions for application, other than saying these COSs are for LMIC settings.

To be flexible and adaptable for a range of LMIC settings, and feasible within the available time and resource constraints, the planned interviews are limited in number and relatively short and semistructured and the transcription will focus only on the most relevant sections. Due to internationally different clinical standards and the potential stigma associated with labelling people as 'at risk', we decided not to conduct interviews focusing on prevention only. This may risk missing outcomes which would be obtained from a more in-depth exploration.

Although our aim is to secure participation from a wide spread of geographical areas, there are a number of challenges to involvement. To take part in Delphi surveys and consensus meetings, English proficiency, a computer and internet access will be required. This poses challenges for engagement with some stakeholder groups, inclusiveness and participant attrition. Additionally, since patient and public involvement is still a developing concept in many contexts and specifically in global health research,[40–43] identifying appropriate patient representatives for our consensus panel may prove challenging.

Despite these challenges, this is the first attempt to develop COSs for multimorbidity seeking input from a wide range of LMIC partners and participants.[21] Additionally, the involvement of various experts in COS and multimorbidity strengthens the COS development process.

Our COS development follows a systematic, widely practiced approach to generate core outcomes and meets the methodological standards put forth by the COMET initiative. It will generate a template for steps in developing international consensus involving LMICs, and two COSs for use in future multimorbidity prevention and treatment trials, helping to generate evidence to address this important global health priority.

**Author affiliations**
[1]School of Health Sciences, University of Dundee, Dundee, UK
[2]Department of Health Sciences, University of York, York, UK
[3]Institute of Psychiatry, Rawalpindi Medical University, Rawalpindi, Pakistan
[4]Centre for Biostatistics, The University of Manchester, Manchester, UK
[5]Florence Nightingale Faculty of Nursing, Midwifery and Palliative Care, King's College London, London, UK
[6]Department of Family Medicine, McMaster University, Hamilton, Ontario, Canada
[7]WHO Collaborating Centre for Research and Training in Mental Health and Service Evaluation Department of Neurosciences, Biomedicine and Movement Sciences, Section of Psychiatry, University of Verona, Verona, Italy
[8]Cochrane Global Mental Health, University of Verona, Verona, Italy
[9]Global Alliance for Chronic Diseases, London, UK
[10]Centre for Reviews and Dissemination and Cochrane Common Mental Disorders, University of York, York, UK
[11]Facultad de Medicina Humana, Centro de Investigación del Envejecimiento (CIEN), Universidad San Martin de Porres, Lima, Peru
[12]Asociación Benéfica PRISMA, Lima, Peru
[13]UCL Respiratory, University College London, London, UK
[14]Chronic Disease Initiative for Africa and Division of Endocrinology, Department of Medicine, University of Cape Town, Cape Town, South Africa
[15]Department of Family Medicine and Population Health, University of Antwerp, Antwerpen, Belgium
[16]Hull York Medical School, York, UK
[17]Department of Population Health, NYU Grossman School of Medicine, New York University, New York, New York, USA
[18]Leeds Institute of Health Sciences, University of Leeds, Leeds, UK
[19]Centre for Reviews and Dissemination, University of York, York, UK

**Acknowledgements** The authors thank Job van Boven (University Medical Center Groningen, University of Groningen, The Netherlands) and Meena Daivadanam (Department of Food Studies, Nutrition and Dietetics, Uppsala University, Sweden) for comments on earlier drafts of this protocol.

**Contributors** NS and JRB conceived the study. JRB, JJK, RZR, LR and NS led the design of the research. JRB, NS and RZR drafted the manuscript. JRB, RZR, JJK, LR, GA, CB, AC, RC, OF-F, JRH, NL, JvO, MP, KS, EPU, RV, JMW, KW, GAZ, NS made substantial contributions to the conception or design of the work. JRB, RZR, JJK, LR, GA, CB, AC, RC, OF-F, JRH, NL, JvO, MP, KS, EPU, RV, JMW, KW, GAZ, NS revised it critically for important intellectual content. JRB, RZR, JJK, LR, GA, CB, AC, RC,

OF-F, JRH, NL, JvO, MP, KS, EPU, RV, JMW, KW, GAZ, NS gave their final approval of the version to be published. JRB, RZR, JJK, LR, GA, CB, AC, RC, OF-F, JRH, NL, JvO, MP, KS, EPU, RV, JMW, KW, GAZ, NS agree to be accountable for all aspects of the work in ensuring that questions related to the accuracy or integrity of any part of the work are appropriately investigated and resolved.

**Funding** This research was funded by the National Institute for Health Research (NIHR) (17/63/130) using UK aid from the UK Government to support global health research. The views expressed in this publication are those of the author(s) and not necessarily those of the NIHR or the UK government. The research is also receiving support from Bradford District Care NHS Foundation Trust. Oscar Flores-Flores is supported by the Research Training in Chronic, non-communicable respiratory diseases in Peru, Fogarty International Center, US National Institutes of Health (D43TW011502). The core outcome set development is also supported by the Global Alliance for Chronic Diseases (GACD) Multimorbidity Working Group; the World Psychiatry Association; and by Cochrane Common Mental Disorders which is funded by NIHR (NIHR 129457) and Cochrane Global Mental Health (https://globalmentalhealth.cochrane.org). It is supported by the NCD Alliance by dissemination of participation opportunities to people living with multiple health conditions in its network.

**Competing interests** None declared.

**Patient consent for publication** Not applicable.

**Provenance and peer review** Not commissioned; externally peer reviewed.

**ORCID iDs**
Jan R. Boehnke http://orcid.org/0000-0003-0249-1870
Jamie J. Kirkham http://orcid.org/0000-0003-2579-9325
Louise Rose http://orcid.org/0000-0003-1700-3972
Gina Agarwal http://orcid.org/0000-0002-5691-4675
John R. Hurst http://orcid.org/0000-0002-7246-6040
Kamran Siddiqi http://orcid.org/0000-0003-1529-7778
Rajesh Vedanthan http://orcid.org/0000-0001-7138-2382
Judy Wright http://orcid.org/0000-0002-5239-0173
Najma Siddiqi http://orcid.org/0000-0003-1794-2152

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
