## [Reviewer comments · BMJ Open]

ARTICLE DETAILS

TITLE (PROVISIONAL)	Development of a core outcome set for multimorbidity trials in low- and middle-income countries (COSMOS): Study Protocol
AUTHORS	Boehnke, Jan Rasmus; Rana, Rusham Zahra; Kirkham, Jamie J.; Rose, Louise; Agarwal, Gina; Barbui, Corrado; Chase, Alyssa; Churchill, Rachel; Flores-Flores, Oscar; Hurst, John; Levitt, Naomi; van Olmen, Josefien; Purgato, Marianna; Siddiqi, Kamran; Uphoff, Eleonora; Vedanthan, Rajesh; Wright, Judy; Wright, Kath; Zavala, Gerardo A.; Siddiqi, Najma

VERSION 1 – REVIEW

REVIEWER	Lehmacher, Walter University of Cologne, Medical Statistics
REVIEW RETURNED	19-Jun-2021

GENERAL COMMENTS	no
----

REVIEWER	Butcher, Nancy Sick Kids, Child Health Evaluative Sciences
REVIEW RETURNED	29-Jun-2021

GENERAL COMMENTS	The authors have described their plans to develop a COS for multimorbidity trials in low- and middle-income countries, an important initiative that will help improve the usefulness and impact of future trials in this area. In writing this protocol, the authors have followed the COS-STAP reporting guideline and considered the COS-STAD standards in the development of their methods. The authors also describe involving patients and families in the development of their COS to ensure outcomes are relevant to the population being studied. The protocol would benefit from some further clarifications of terms and some modifications to text to ensure that the methods are clear, as described below. This review was completed by Dr. Nancy Butcher and Katy Goren from the Hospital for Sick Children. Abstract 1) It is unclear to the reader, as written, whether the three stages of COS development will be completed for the two COS together or separately, please clarify here. 2) Consider specifying who will be included in the consensus meeting as this will have a large impact on the ultimate COS (page 4, line 38). Introduction Major
---

3) Given the presence of the existing Cochrane review and COS for multimorbidity in adults, please provide a stronger rationale for creating a COS for multimorbidity that is specific to LMIC. For example, what types of differences might be expected in endorsed outcomes and outcome measurement instruments in a LMIC?
4) Additionally, if the authors' aim is to bridge the gap between past research and your current work have the authors considered incorporating previous findings/outcomes into your COS development process instead of starting from the beginning (pg. 7, line 7)?
5) Please describe the rationale behind creating two separate COS for prevention and treatment and why each is important.

Minor

6) It would be helpful if a definition for chronic non-communicable diseases (page 6, line 19) was provided in addition to the definition chronic communication diseases.
7) Please consider expanding on the differences between disease patterns/healthcare systems in LMIC compared to HIC (page 6, line, 55).
8) Additionally, please describe the context of use for which each COS is to be applied in the introduction (COS-STAP Criteria 3c).

Aims and objectives

9) It would be helpful to provide a distinction between what is meant by simple vs complex interventions, for example, by adding definitions or examples (pg. 7, line 17).
10) The operationalization of "at risk of" multimorbidity in the context of this COS (pg. 7, line 11) is strong recommended, as having a clear definition of the health condition being studied is necessary as per COS-STAD – Standard 2. For example, does this mean having at least one documented disease risk factor such as age, family history, genetics, etc.?

Methods

12) Good work registering the protocol on COMET and clearly stating that the protocol was developed with reference to the COS-STAP, and that COS development process will follow the COS-STAD recommendations. It is unclear, however, if the development process will be completed in duplicate for each COS or if each step will only be conducted once to create both COS (pg. 7, line 46).

1. Outcome list generation

13) Please clarify whether it is healthcare professionals with experience treating those with multimorbidity and/or at risk of multimorbidity in the generation of the initial list of outcomes (as per COS-STAD Standard 8).

1A Systematic review

14) The authors have followed recommended methods by the COMET initiative by conducting a systematic review as the first stage in generating the initial list of outcomes. Using a PRISMA 2020 diagram for your search, and separating papers that report on prevention, treatment, and both, may be helpful.

Search methods

15) Did the authors have an information specialist review their search strategy using the Peer Review for Electronic Search Strategies (PRESS)?

Selection criteria

	16) Consider reframing the selection criteria using the population, intervention, comparator, outcome template to make it clearer. 17) Should the 2019 World Bank Definitions be cited? (pg. 8, line 28). Study selection 18) Please describe the process of removing duplicates. Data collection 19) Please describe what tool you will be using to extract data (e.g., Excel, REDCap), if known 20) Please describe if all outcomes identified at this stage will be kept or if outcomes will be dropped or modified. 21) Please clarify what is meant by outcomes will be grouped by combination of condition (page 9, line 12). 1B interviews 22) Please clarify what each group means where it is written that we anticipate 15 to 20 participants from each group (pg. 9, line 40). It is not clear if a group is defined by age, LMIC, or type of care used. 23) Please describe who will be conducting the interviews and if it is required that they are experienced qualitative facilitators. 24) Specify who will be responsible for integrating interview outcome findings with systematic review findings and how outcomes will be included/added/dropped as per COS-STAD Standard 10. 25) The relevance of collecting marital status is unclear to the aim of the interviews. 2. Delphi surveys with stakeholders The authors are planning to bring together various stakeholder groups for the Delphi process, which is a clear strength of their methods. 26) Please specify if the healthcare workers included will have experience working with patients at risk of multimorbidity or with multimorbidity (COS-STAD Standard 6). 2A. Round One The authors should be praised in that they plan to use advice from the community and public representatives to ensure outcomes are in lay terms. The piloting of the survey prior to distribution is also appreciated as a methodological strength. 27) Please describe if participants will be sent both questionnaires at the same time or if they will complete the questionnaires separately for the two COS in the Delphi rounds. The authors have predefined the scoring process and consensus definitions in accordance with COS-STAD recommendations. It is appreciated that they have noted that they will provide participants with the opportunity to add additional outcomes and other comments in Round 1. 28) It would be helpful if the authors could clarify if or how partially completed surveys will be used in the analysis, as they have done for Round Two. 3. Outcome consensus meetings 29) Two separate consensus meetings are noted; it would be helpful to note if these meetings will happen at different time points and if participants can choose to attend one meeting but not the other.
--	---

- 30) It is important to provide a clearer consensus definition for including and excluding outcomes (pg. 12, line 24).
- 31) How many participants will be invited to the consensus meeting to have a fulsome discussion virtually?
- 32) Do the authors have any planned mitigation situations for common situation(s) in COS development where the Delphi fails to reduce the number of outcomes to a manageable number for discussion at the consensus meeting and/or a manageable number for implementation following the consensus meeting?
- 33) The provision of additional support to individuals living with multimorbidity and their caregivers before the meeting is noted as a strength of their patient engagement approach. The authors could additionally specify whether/how ambiguity of language in the outcomes will be avoided (eg through consultations or pilot testing with the public) via (COS-STAD Standard 11).

Dissemination

A noted strength is the provision of lay summaries including infographics in relevant languages for community and public audiences.

Discussion

We agree with the noted strengths of their COS development approaches eg, including individuals and their caregivers living with multimorbidity is needed to determine what outcomes are important to this population. Additionally, the involvement of various experts in COS development and multimorbidity is of great value to the COS development process.

34) Given there is already a COS developed for multimorbidity that is not specific to LMIC, clearly defining what regions/populations this COS can be used needs to be clear so that it can be successfully implemented in future research in the correct settings.

35) Please describe how you plan to deal with the limitation of generalizability, such as defining the regions that your research team believes that the COS can be applied.

36) Please clarify in your third limitation what you mean by “we decided not to conduct interviews on prevention only” (pg. 14, line 48). If interviews are not being conducted for treatment, this may have the consequence of limiting the extent of patient engagement of the COS and potentially its relevance (Please see COS-STAD Standard 8).

37) A further point of consideration is reporting on the measurement tools/definitions used to measure outcomes included in both final COS. A lack of recommended outcome measurement instruments is a common barrier to the implementation of COS in practice. Do the authors have plans to develop recommendations for how to measure the selected outcomes

Conflicts of interest

38) Please describe any conflicts of interest, and if no conflicts of interest are present, please specify no conflicts of interest as per COS-STAP Criteria 13

Additional minor suggestions

39) Please go over the references throughout the entire document as references appear to be lacking in some areas (i.e., add a reference for every sentence in which the text is not an original thought).

	40) There is inconsistency in acronyms usage throughout the paper. Please ensure if a term is used once and defined as an acronym, all following terms should be in acronym form (i.e., low- and middle- income countries pg. 8, line 26). Please define acronyms before using them to help the reader (i.e., HIV, page 6, line 19; COSMOS pg. 7, line 36). 41) Please do not use short-form language (i.e., incl. page 9, line 5).
REVIEWER	Ekdahl, Anne Karolinska Institutet Department of Neurobiology Care Sciences and Society, Section of Clinical geriatrics
REVIEW RETURNED	23-Aug-2021
GENERAL COMMENTS	Dear authors, I find this study protocol very interesting, and as far as I can see, you have followed the adequate checklist. I miss a reference that might be of value: Akpan A et al., R Standard set of health outcome measures for older persons. BMC Geriatrics. 2018;18(1):36 as these outcomes are relevant for the populations which are going to be studied

VERSION 1 – AUTHOR RESPONSE

Reviewer: 1 Dr. Walter Lehmacher, University of Cologne	
Comments to the Author: No	No comments provided.
Reviewer: 2 Dr. Nancy Butcher, Sick Kids	
The authors have described their plans to develop a COS for multimorbidity trials in low- and middle-income countries, an important initiative that will help improve the usefulness and impact of future trials in this area. In writing this protocol, the authors have followed the COS-STAP reporting guideline and considered the COS-STAD standards in the development of their methods. The authors also describe involving patients and families in the development of their COS to ensure outcomes are relevant to the population being studied. The protocol would benefit from some further clarifications of terms and some modifications to text to ensure that the methods are clear, as described below.	Thank you for the positive comments about the purpose and methods of the study, including involvement of patients and families. The clarifications requested are given below.
Abstract 1) It is unclear to the reader, as written, whether the three stages of COS development will be completed for the two COS together or separately, please clarify here.	We have added this detail (page 4): ‘the development of these two COS will occur together in parallel in three stages’

2) Consider specifying who will be included in the consensus meeting as this will have a large impact on the ultimate COS (page 4, line 38).	We have added this information: ‘consensus meetings including all of these stakeholders will be held to discuss the Delphi survey results and agree outcomes for inclusion in the two COS’ (page 4)
Introduction 3) Given the presence of the existing Cochrane review and COS for multimorbidity in adults, please provide a stronger rationale for creating a COS for multimorbidity that is specific to LMIC. For example, what types of differences might be expected in endorsed outcomes and outcome measurement instruments in a LMIC?	The rationale for the COS is given in the introduction (page 6). We have added examples of potential differences (page 7)
4) Additionally, if the authors’ aim is to bridge the gap between past research and your current work have the authors considered incorporating previous findings/outcomes into your COS development process instead of starting from the beginning (pg. 7, line 7)?	This was considered, but would have been a less rigorous approach. The literature review in the previous work was more limited. We were aiming to develop a COS specifically for LMIC, rather than an all-encompassing global COS covering both LMIC and HIC. Additionally, the previous COS development focused only on treatment studies. Therefore, it was not appropriate to incorporate the previous COS, although we will of course note which outcomes are common to both.
5) Please describe the rationale behind creating two separate COS for prevention and treatment and why each is important.	We have added this to the introduction (page 7) ‘Separate COS are needed given the often differing targets for prevention and treatment interventions’.
6) It would be helpful if a definition for chronic non-communicable diseases (page 6, line 19) was provided in addition to the definition chronic communication diseases.	This has been added (page 6) ‘Non-communicable diseases are chronic conditions that cannot be directly transmitted between people

	e.g. diabetes, heart disease and chronic lung disease’.
7) Please consider expanding on the differences between disease patterns/healthcare systems in LMIC compared to HIC (page 6, line, 55).	We have added differences: ‘differences in severity of disease symptoms or complications and in healthcare availability and financing may influence outcome priorities’ (page 6) Further expansion would be difficult within the word count limits, and we think that is not the main focus of the paper.
8) Additionally, please describe the context of use for which each COS is to be applied in the introduction (COS-STAP Criteria 3c).	The introduction describes the context for use of the 2 COS: ‘trials on the prevention and treatment of multimorbidity in an LMIC setting’ (page 7)
Aims and objectives 9) It would be helpful to provide a distinction between what is meant by simple vs complex interventions, for example, by adding definitions or examples (pg. 7, line 17).	We have added a definition of complex interventions ‘Complex interventions are those with multiple interacting components (e.g health risk behaviour change interventions), which contrast with ‘simple’ interventions, such as pharmacological interventions.’ (page 7)
10) The operationalization of “at risk of” multimorbidity in the context of this COS (pg. 7, line 11) is strong recommended, as having a clear definition of the health condition being studied is necessary as per COS-STAD – Standard 2. For example, does this mean having at least one documented disease risk factor such as age, family history, genetics, etc.?	Since all people could be ‘at risk’ of multimorbidity, and the list of potential risk factors would be extremely long, we do not think that this should be defined further. Multimorbidity has been defined, and age above 18 years. This clearly identifies the target population.
Methods 12) Good work registering the protocol on COMET and clearly stating that the protocol was developed with reference to the COS-STAP, and that COS development process will follow the COS-STAD recommendations. It is unclear, however, if the development process will be completed in duplicate for each	The Delphi and the consensus meetings will be conducted separately for each COS.

COS or if each step will only be conducted once to create both COS (pg. 7, line 46).	See page 10: We will conduct separate Delphi surveys for the prevention and treatment COS and page 12: 'the final step in the consensus process will be two online consensus meetings: one to agree the COS for prevention of multimorbidity and one for the COS for treatment of multimorbidity'
1 Outcome list generation 13) Please clarify whether it is healthcare professionals with experience treating those with multimorbidity and/or at risk of multimorbidity in the generation of the initial list of outcomes (as per COS-STAD Standard 8).	Healthcare professionals are not involved in generation of the initial list of outcomes. Views of researchers and healthcare professionals usually inform trial design and would therefore already be reflected in the body of published literature in our systematic review.
1A Systematic review 14) The authors have followed recommended methods by the COMET initiative by conducting a systematic review as the first stage in generating the initial list of outcomes. Using a PRISMA 2020 diagram for your search, and separating papers that report on prevention, treatment, and both, may be helpful.	This will be part of the systematic review paper when it is written up for publication.
Search methods 15) Did the authors have an information specialist review their search strategy using the Peer Review for Electronic Search Strategies (PRESS)?	Co-author Judy Wright is an Information Specialist with extensive experience of constructing and executing searches, including searching for global health literature. We did not use PRESS.
Selection criteria 16) Consider reframing the selection criteria using the population, intervention, comparator, outcome template to make it clearer.	The section on selection criteria already conforms to PICO, describing the population, followed by interventions and controls (any comparators). The outcomes are not defined as the aim of the review is to gather information on all outcomes reported in the included trials.
17) Should the 2019 World Bank Definitions be cited? (pg. 8, line 28).	This has been added (REF29).
Study selection 18) Please describe the process of removing duplicates.	Duplicates identified in Endnote were removed. We have not included this level of detail in the paper given the review

	forms only one step in a multi-step process, but can add it if the editor advises this is required.
Data collection 19) Please describe what tool you will be using to extract data (e.g., Excel, REDCap), if known	We are using Excel, but as above, think this level of detail may not be needed in a paper such as this.
20) Please describe if all outcomes identified at this stage will be kept or if outcomes will be dropped or modified.	All outcomes identified at this stage will be kept, as described on page 11, but we have clarified that non-disease specific outcomes will be removed. “We will ensure the outcomes that enter the Delphi are non-disease specific (important for the scope of this COS), have relevance to LMIC, and avoid duplication.”
21) Please clarify what is meant by outcomes will be grouped by combination of conditions (page 9, line 12).	This means the combination of chronic conditions that were the focus of the trial from which the outcomes have been extracted (added on page 8, “by combination of conditions that the trial targeted”)
1B interviews 22) Please clarify what each group means where it is written that we anticipate 15 to 20 participants from each group (pg. 9, line 40). It is not clear if a group is defined by age, LMIC, or type of care used.	The groups are defined in the previous 2 sentences on page 9: age (over/under 65 years); sex (male/female); using community or primary care versus secondary/specialist healthcare services; and region (World Bank income groups: low, lower middle, upper middle(29)). We will include a range of LMICs in diverse geographic locations.
23) Please describe who will be conducting the interviews and if it is required that they are experienced qualitative facilitators.	We have added: ‘Interviews will be conducted by researchers with experience of conducting qualitative interviews.’ (page 10)
24) Specify who will be responsible for integrating interview outcome findings with systematic review findings and how outcomes will be included/added/dropped as per COS-STAD Standard 10.	As already mentioned above, no outcomes will be dropped at this stage. The research team will collate the list of outcomes generated from the review and interviews.

25) The relevance of collecting marital status is unclear to the aim of the interviews.	This is to ensure a diverse group of respondents.
2. Delphi surveys with stakeholders The authors are planning to bring together various stakeholder groups for the Delphi process, which is a clear strength of their methods.	Thank you.
26) Please specify if the healthcare workers included will have experience working with patients at risk of multimorbidity or with multimorbidity (COS-STAD Standard 6).	Yes, the selection of healthcare workers requires experience working with multimorbidity. Added on page 10: "... healthcare professionals experienced in working with patients with multimorbidity;"
2A. Round One The authors should be praised in that they plan to use advice from the community and public representatives to ensure outcomes are in lay terms. The piloting of the survey prior to distribution is also appreciated as a methodological strength.	Thank you.
27) Please describe if participants will be sent both questionnaires at the same time or if they will complete the questionnaires separately for the two COS in the Delphi rounds.	They will be sent both questionnaires at the same time.
The authors have predefined the scoring process and consensus definitions in accordance with COS-STAD recommendations. It is appreciated that they have noted that they will provide participants with the opportunity to add additional outcomes and other comments in Round 1.	Thanks.
28) It would be helpful if the authors could clarify if or how partially completed surveys will be used in the analysis, as they have done for Round Two.	We have added this information on page 11. "All responses, including from partially completed questionnaires will be used in analyses."
3. Outcome consensus meetings 29) Two separate consensus meetings are noted; it would be helpful to note if these meetings will happen at different time points and if participants can choose to attend one meeting but not the other.	There will be separate meetings. The time points and details of whether participants can join both will need to be confirmed nearer the time, depending on what is feasible timetabling-wise.
30) It is important to provide a clearer consensus definition for including and excluding outcomes (pg. 12, line 24).	We have been advised by experts in COS development, (who have also

	been part of the COMET group developing methods) that ranking by consensus of group members, as described in the paper, is the recommended approach, rather than setting a priori criteria for inclusion/exclusion of outcomes.
31) How many participants will be invited to the consensus meeting to have a fulsome discussion virtually?	It is difficult to set the exact number of participants a priori, as it will depend on the response to the Delphi. The important goal is to achieve balanced representation across stakeholder groups (as already specified in the protocol). Members of the team (LR and JK) have considerable experience of running consensus meetings, including online meetings, and have found it possible to manage discussions even with large numbers of participants by using functions such as break out rooms, a series of smaller meetings and chat. We have not specified the participant numbers, but have added more details about how meaningful discussions will be supported (Pages 12 & 13)
32) Do the authors have any planned mitigation situations for common situation(s) in COS development where the Delphi fails to reduce the number of outcomes to a manageable number for discussion at the consensus meeting and/or a manageable number for implementation following the consensus meeting?	Mitigation strategies will need to be adaptive, depending on the number of outcomes identified for the Delphi, the number that meet criteria for discussion in the consensus groups and the number proposed to be included in the COS by the group. The research team will ensure the outcomes that enter the Delphi are non-disease specific (important for the scope of this COS), have relevance to LMIC, and avoid duplication. All such outcomes will be discussed at the consensus meetings. Those scoring below the specified

	thresholds may be excluded quite quickly, and the remainder ranked. If consensus cannot be reached on all outcomes or there are outstanding issues to consider, we will convene separate meetings with the stakeholder members until a final COS is ratified. We have added this information on page 11 and on page 12/13.
33) The provision of additional support to individuals living with multimorbidity and their caregivers before the meeting is noted as a strength of their patient engagement approach. The authors could additionally specify whether/how ambiguity of language in the outcomes will be avoided (eg through consultations or pilot testing with the public) via (COS-STAD Standard 11).	Piloting with the public is not possible within time and resource limits. However, researchers at each site will check the interview questions for ambiguity, and translation and back translation of the questions will help to improve mutual understanding of the questions, and their cultural relevance and appropriateness. This process has been described on page 10.
Dissemination A noted strength is the provision of lay summaries including infographics in relevant languages for community and public audiences.	Thank you.
Discussion We agree with the noted strengths of their COS development approaches eg, including individuals and their caregivers living with multimorbidity is needed to determine what outcomes are important to this population. Additionally, the involvement of various experts in COS development and multimorbidity is of great value to the COS development process.	Thank you. We have added this to strengths. (Page 15)
34) Given there is already a COS developed for multimorbidity that is not specific to LMIC, clearly defining what regions/populations this COS can be used needs to be clear so that it can be successfully implemented in future research in the correct settings.	This has already been defined in the aims. We have not defined any specific regions- see response below.
35) Please describe how you plan to deal with the limitation of generalizability, such as defining the regions that your research team believes that the COS can be applied.	We have acknowledged there may be potential limits to generalisability as it is not possible to have representation from all LMICs in the

	consensus process; nevertheless we included literature from all LMIC in generating the list of outcomes, and the diverse range of countries represented means it is likely that the outcomes selected will be those that are common across countries and cultures. We do not therefore intend to define or limit regions for application, other than saying these COS are for LMIC settings. We have now included this explanation in the Discussion on page 15.
36) Please clarify in your third limitation what you mean by “we decided not to conduct interviews on prevention only” (pg. 14, line 48). If interviews are not being conducted for treatment, this may have the consequence of limiting the extent of patient engagement of the COS and potentially its relevance (Please see COS-STAD Standard 8).	We conducted interviews with people with multimorbidity and asked them about prevention outcomes as well as treatment outcomes. The outcomes elicited should therefore be relevant to both prevention and treatment COS. We did not conduct interviews in those only at risk of multimorbidity (i.e. not currently experiencing multimorbidity). Apart from the resource limitations guiding this decision, to do so would have involved labelling people as ‘at risk’ of multimorbidity, which we did not think was justifiable, as we could get the required information from interviews with people with established multimorbidity diagnoses.
37) A further point of consideration is reporting on the measurement tools/definitions used to measure outcomes included in both final COS. A lack of recommended outcome measurement instruments is a common barrier to the implementation of COS in practice. Do the authors have plans to develop recommendations for how to measure the selected outcomes	We plan to report the measures used to assess each outcome in the included trials in our literature review. However, recommendation of measures is not within the scope of this work; it will be undertaken if we are able to secure further funding in future.
Conflicts of interest	Conflicts of interest have been declared in the submission, but as we submitted through medRxiv this

38) Please describe any conflicts of interest, and if no conflicts of interest are present, please specify no conflicts of interest as per COS-STAP Criteria 13	may not have been visible to reviewers. Now added on the title page.
Additional minor suggestions 39) Please go over the references throughout the entire document as references appear to be lacking in some areas (i.e., add a reference for every sentence in which the text is not an original thought).	References have been checked and updated, as needed.
40) There is inconsistency in acronyms usage throughout the paper. Please ensure if a term is used once and defined as an acronym, all following terms should be in acronym form (i.e., low- and middle- income countries pg. 8, line 26). Please define acronyms before using them to help the reader (i.e., HIV, page 6, line 19; COSMOS pg. 7, line 36).	We have reviewed and amended as needed for consistency in use of abbreviations and acronyms. HIV is a commonly used abbreviation, included as a word in Merriam-Webster's dictionary, so we have not spelled it out.
41) Please do not use short-form language (i.e., incl. page 9, line 5).	This has been corrected.
Reviewer: 3 Dr. Anne Ekdahl, Karolinska Institutet Department of Neurobiology Care Sciences and Society, Lunds Universitet Institutionen for kliniska vetenskaper i Lund	
I find this study protocol very interesting, and as far as I can see, you have followed the adequate checklist. I miss a reference that might be of value: Akpan A et al., R Standard set of health outcome measures for older persons. BMC Geriatrics. 2018;18(1):36 as these outcomes are relevant for the populations which are going to be studied	Thank you for your review and positive comments. Thank you also for sharing this interesting reference. On balance we have decided not to include it, as although it describes development of a COS that would be relevant to a subgroup of our population, it would not be applicable to the wider target population.

VERSION 2 – REVIEW

REVIEWER	Butcher, Nancy Sick Kids, Child Health Evaluative Sciences
REVIEW RETURNED	17-Nov-2021
GENERAL COMMENTS	Thank you for the thorough and thoughtful responses as well as the revision of your protocol. One final note would be to consider providing some additional methodological details as to how the outcome ranking during the consensus process will be performed for a prior transparency of methods regarding the final selection of the included outcomes. Good luck with your COS.